# Codon bias imposes a targetable limitation on *KRAS*-driven therapeutic resistance

Moiez Ali[1,*], Erin Kaltenbrun[1,*], Gray R. Anderson[1], Sarah Jo Stephens[1], Sabrina Arena[2,3], Alberto Bardelli[2,3], Christopher M. Counter[1,**] & Kris C. Wood[1,**]

*KRAS* mutations drive resistance to targeted therapies, including EGFR inhibitors in colorectal cancer (CRC). Through genetic screens, we unexpectedly find that mutant *HRAS*, which is rarely found in CRC, is a stronger driver of resistance than mutant *KRAS*. This difference is ascribed to common codon bias in *HRAS*, which leads to much higher protein expression, and implies that the inherent poor expression of *KRAS* due to rare codons must be surmounted during drug resistance. In agreement, we demonstrate that primary resistance to cetuximab is dependent upon both *KRAS* mutational status and protein expression level, and acquired resistance is often associated with *KRAS*[Q61] mutations that function even when protein expression is low. Finally, cancer cells upregulate translation to facilitate *KRAS*[G12]-driven acquired resistance, resulting in hypersensitivity to translational inhibitors. These findings demonstrate that codon bias plays a critical role in *KRAS*-driven resistance and provide a rationale for targeting translation to overcome resistance.

[1] Department of Pharmacology and Cancer Biology, Duke University, Durham, North California 27710, USA. [2] Department of Oncology, University of Torino, 10060 Candiolo, Torino, Italy. [3] Candiolo Cancer Institute-FPO, IRCCS, 10060 Candiolo, Torino, Italy. * These authors contributed equally to this work. ** These authors jointly supervised this work. Correspondence and requests for materials should be addressed to C.M.C. (email: chris.counter@duke.edu) or to K.C.W. (email: kris.wood@duke.edu).

Primary and acquired resistance to targeted therapies place major limitations on the clinical efficacy of these drugs, despite their promising preclinical potential[1]. One proposed method to overcome resistance has been to identify genetic[2], epigenetic and signalling alterations that underlie sensitivity and resistance, and then use these as biomarkers to better select patients that will benefit from specific mono- or combination therapies[3–5]. In this regard, numerous studies have specifically implicated activating mutations in *KRAS* as a major driver of both primary and acquired resistance to diverse targeted therapies.

In the area of primary resistance, activating mutations in *KRAS* track with poor responses to EGFR inhibition with the monoclonal antibodies (mAbs) cetuximab and panitumumab, or small molecule EGFR inhibitors like gefitinib and erlotinib, in colorectal cancer (CRC) and non-small cell lung cancer (NSCLC) patients[6–10]. Activating mutations in *KRAS* have also been proposed as a mechanism of primary resistance to the tyrosine kinase inhibitor (TKI) imatinib in *KIT*- or *PDGFRA*-mutant gastrointestinal stromal tumours[11] and to JAK inhibitors in *JAK2*-mutant myeloproliferative neoplasms[12].

In addition to cases of primary resistance, the emergence of *de novo KRAS* mutations has also been linked to acquired resistance in multiple cancer types and contexts: to both anti-EGFR therapy and MEK1/2 inhibitors in CRC[13,14], to imatinib in chronic myelogenous leukaemia[15], and to BRAF/MEK inhibitors in melanoma[16]. In CRC, despite achieving initial responses, patients who originally present with no detectable mutations in *KRAS* (wild-type (WT) *KRAS*, or *KRAS^{WT}*, disease) develop acquired resistance to anti-EGFR therapy that is commonly driven by *KRAS* mutations, limiting the clinical benefit of this therapy[13,17–19]. Curiously, the mutations detected in this setting of acquired resistance are a balance of G12/G13 and Q61 mutations, the latter of which are rarely found in treatment naive CRC[20,21]. An improved understanding of the biology and signalling that support *KRAS*-mediated resistance may therefore give rise to new therapeutic strategies for these refractory tumours.

*KRAS* belongs to a family of three genes, the other two being *HRAS* and *NRAS*. Interestingly, it is *KRAS* that is the most commonly mutated of the three in a wide spectrum of cancers and in the setting of resistance[22]. Despite this apparent contrast in epidemiological data, the encoded proteins are very similar, and in fact share ∼85% sequence identity[22]. However, we discovered that the coding nucleotide sequence varies extensively between these three genes. Specifically, *HRAS* is enriched in common codons that yield robust translation and hence high protein expression. *KRAS* is characterized by rare codons, yielding poor translation and low expression, while *NRAS* has a mixture of common and rare codons and intermediate expression[22].

Here, we show that this rare-codon bias, entrenched in the nucleotide sequence of *KRAS*, plays a critical role in both primary and acquired resistance and may underlie observations of elevated KRAS expression in models of resistance to the mAb cetuximab, the emergence of more rare *KRAS* amino acid site mutations in patients with acquired cetuximab resistance, and provide a novel therapeutic avenue to combat resistance.

## Results

**Mutant *HRAS* confers greater drug resistance than mutant *KRAS*.** Previously, we described a systematic approach to identify novel drug resistance pathways by screening cancer cell lines with a pooled library of 36 mutant complementary DNAs (cDNAs) that constitutively activate or inhibit 17 key oncogenic signalling pathways[23]. Using this approach, we observed that across nine genotype- and lineage-defined groupings of 14 cell lines, treated with a total of 14 targeted therapies, there was a marked differential resistance conferring potential between oncogenic HRAS^{G12V} and KRAS^{G12V} (Fig. 1a). Specifically, oncogenic HRAS^{G12V} conferred resistance in 27 of 29 screens with a broad spectrum of therapeutics using a previously established scoring threshold (Fig. 1b)[23], and it scored as the top overall hit in 22 of 29 of the screens (Fig. 1c). By contrast, KRAS^{G12V} scored only rarely (6 of 29 screens), and never as the top overall hit. Moreover, even in screens where both HRAS^{G12V} and KRAS^{G12V} reached scoring criteria, HRAS^{G12V} consistently achieved higher enrichment scores (the relative abundance of each construct in the presence of drug normalized to the same value in the absence of drug), implying stronger resistance (Fig. 1d). To determine whether the ability to score in our screens correlated with the expression of each construct, we performed western blot analysis using extracts from cell lines in which both constructs (A375) or only HRAS^{G12V} (SKBR3, PC9, NCIH508) scored. In all cases examined, HRAS^{G12V} was detected at higher levels compared to KRAS^{G12V} (Fig. 1e). Thus, the level of RAS expression correlated with resistance, with HRAS consistently expressed at higher levels.

**Codon bias underlies increased resistance conferred by *HRAS*.** One feature of *RAS* genes that could explain the enhanced resistance conferred by *HRAS* relative to *KRAS* is codon bias. To address this hypothesis, we created native-codon and codon-modified cDNAs encoding FLAG-tagged, oncogenic *HRAS* and *KRAS*. Specifically, we created a rare codon-enriched, oncogenic *HRAS^{G12D}* cDNA by converting key common codons to rare codons (termed *HRAS^{R,G12D}*). Reciprocally, we also created a common codon-enriched, oncogenic *KRAS^{G12D}* cDNA by exchanging rare codons for common codons (termed *KRAS^{C,G12D}*). These four constructs as well as an empty vector control were individually stably expressed in three very different human cancer cell lines, namely the *BRAF^{V600E}* mutation-positive melanoma cell line UACC-62 (Fig. 2a–c), the *EGFR* mutation-positive NSCLC cell line PC9 (Fig. 2d–f), and the *PDGFR*-amplified NSCLC cell line NCIH1703 (Fig. 2g–i). In agreement with the known effect of codon bias on *RAS* protein expression[22], oncogenic HRAS^{G12D} was readily detected by immunoblot analysis, and its expression was greatly reduced following the exchange of common codons for rare codons in all three cell lines (Fig. 2a,d,g). Conversely, oncogenic KRAS^{G12D} was very poorly expressed in all three cell lines, an effect that was reversed by changing rare codons to common (Fig. 2a,d,g). When these cell lines were treated with targeted inhibitors against the driver oncogene in each line, namely vemurafenib to inhibit oncogenic BRAF in UACC-62 cells, gefitinib to inhibit oncogenic EGFR in PC9 cells and sunitinib to inhibit PDGFR in NCIH1703 cells, we found that in each case the degree of resistance matched RAS protein expression. Namely, the RAS isoforms with the most common codons (HRAS^{G12D} and KRAS^{C,G12D}) imparted greater resistance than the versions with more rare codons (HRAS^{R,G12D} and KRAS^{G12D}) (Fig. 2b,e,h, quantified in Fig. 2c,f,i). The same was also true in the completely independent assay of anchorage-independent growth (Supplementary Fig. 1). Thus, the ability of oncogenic HRAS to impart resistance to a broad spectrum of cell lines and drugs appears to be due, at least in part, to its high-level protein expression due to its inherent bias towards common codons.

**Codon bias underlies *KRAS*-mediated resistance to cetuximab.** The aforementioned finding appears, at face value, at odds with the clinical observation that oncogenic *KRAS* has been extensively

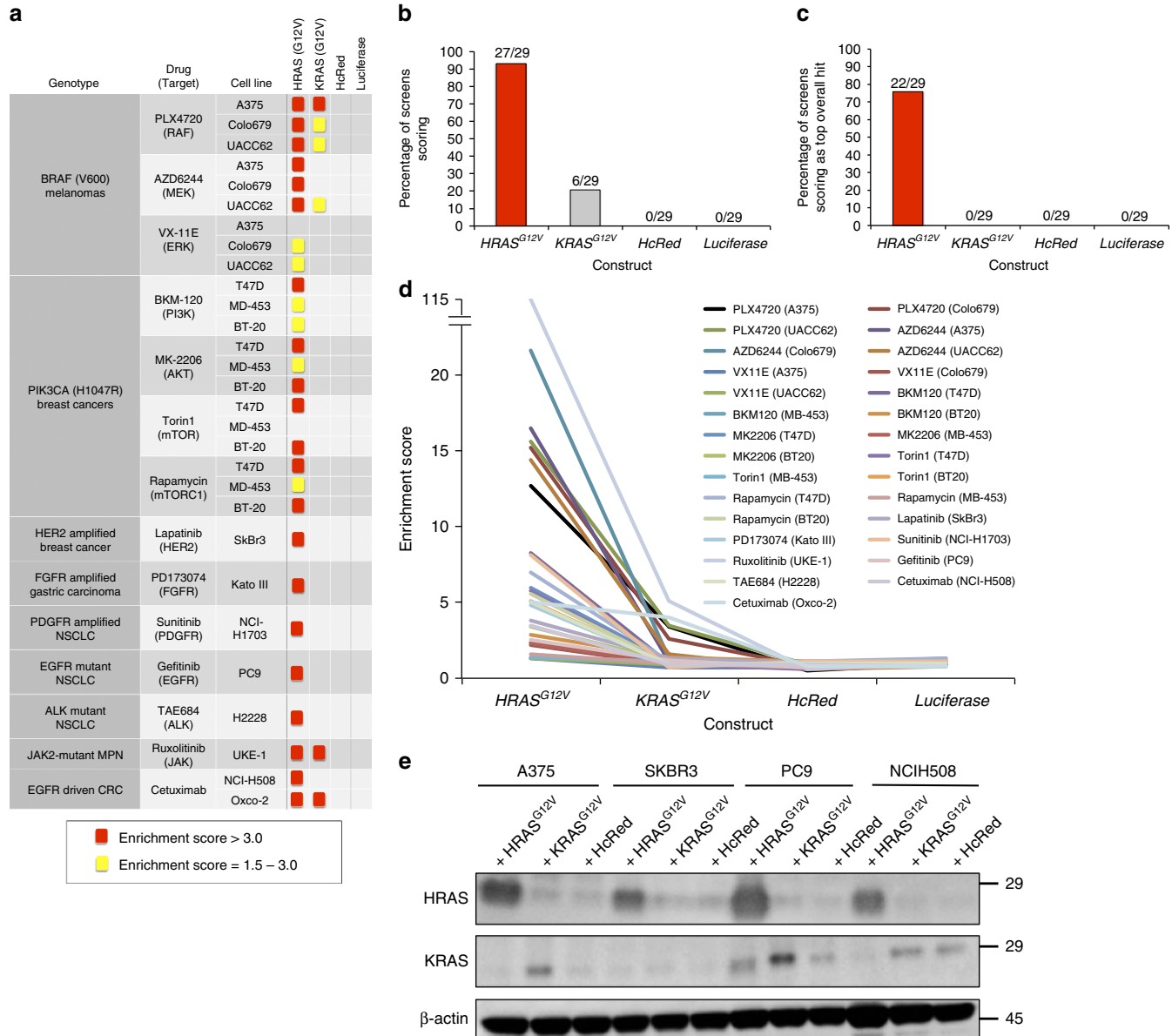

**Figure 1 | Pathway activating screens reveal differential resistance conferring potential between ectopic $HRAS^{G12V}$ and $KRAS^{G12V}$.** (**a**) Enrichment scores for four lentivirally delivered cDNA constructs (encoding $HRAS^{G12V}$, $KRAS^{G12V}$, HcRed and Luciferase) across 29 resistance screens involving diverse kinase inhibitors and cell lines. Enrichment scores reflect the relative abundance of each construct in the presence of drug normalized to the same value in the absence of drug, with scores exceeding 1.5 considered to be indicative of resistance. (**b**) Percentage of screens in **a** in which indicated constructs scored (enrichment score > 1.5). (**c**) Percentage of screens in **a** in which indicated constructs scored as the top overall hit from a library of 40 pathway activating constructs and controls. (**d**) Enrichment score for each construct in each screen. (**e**) Immunoblot analysis with antibodies against HRAS, KRAS or β-actin using extracts isolated from the indicated cell lines stably transduced with a vector encoding the indicated transgene. Images are cropped for clarity.

implicated as a driver of primary and acquired resistance. We posited that perhaps poor expression of KRAS due to rare codons is a barrier to resistance, such that in addition to an oncogenic mutation, resistance may require an increase in KRAS expression. To this end we focused on CRC, a cancer in which oncogenic *KRAS* is well established to be a central driver of primary and acquired resistance to anti-EGFR therapies[13]. The CRC cell line NCIH508, which is $KRAS^{WT}$, sensitive to the drug cetuximab, and can be rendered resistant to cetuximab through expression of oncogenic RAS, was stably transduced with the aforementioned vectors encoding no transgene or $HRAS^{G12D}$, $HRAS^{R,G12D}$, $KRAS^{G12D}$ or $KRAS^{C,G12D}$. As before, expression of the RAS isoforms tracked with codon bias (Fig. 3a). Further, we observed that common codon-enriched *RAS* constructs conferred

resistance to cetuximab in colony growth assays relative to rare codon-enriched *RAS* constructs (Fig. 3b,c). Identical findings were observed upon treatment of cells with an eight-log dilution series of cetuximab, where half-maximal growth inhibition values for both $HRAS^{G12D}$ and $KRAS^{C,G12D}$ were significantly greater than for their rare codon-enriched counterparts (Fig. 3d,e). These results were reproducible in another $KRAS^{WT}$ CRC cell line, LIM1215 (Supplementary Fig. 2A,B). Immunoblots performed in the presence and absence of cetuximab in cells expressing controls or common and rare *RAS* constructs demonstrated on-target inhibition of the mitogen-activated protein kinase (MAPK) pathway, as evidenced by modest to substantial decreases in phospho-ERK (p-ERK) levels in both NCIH508 and LIM1215 cell lines (Fig. 3f, Supplementary Fig. 2C). This

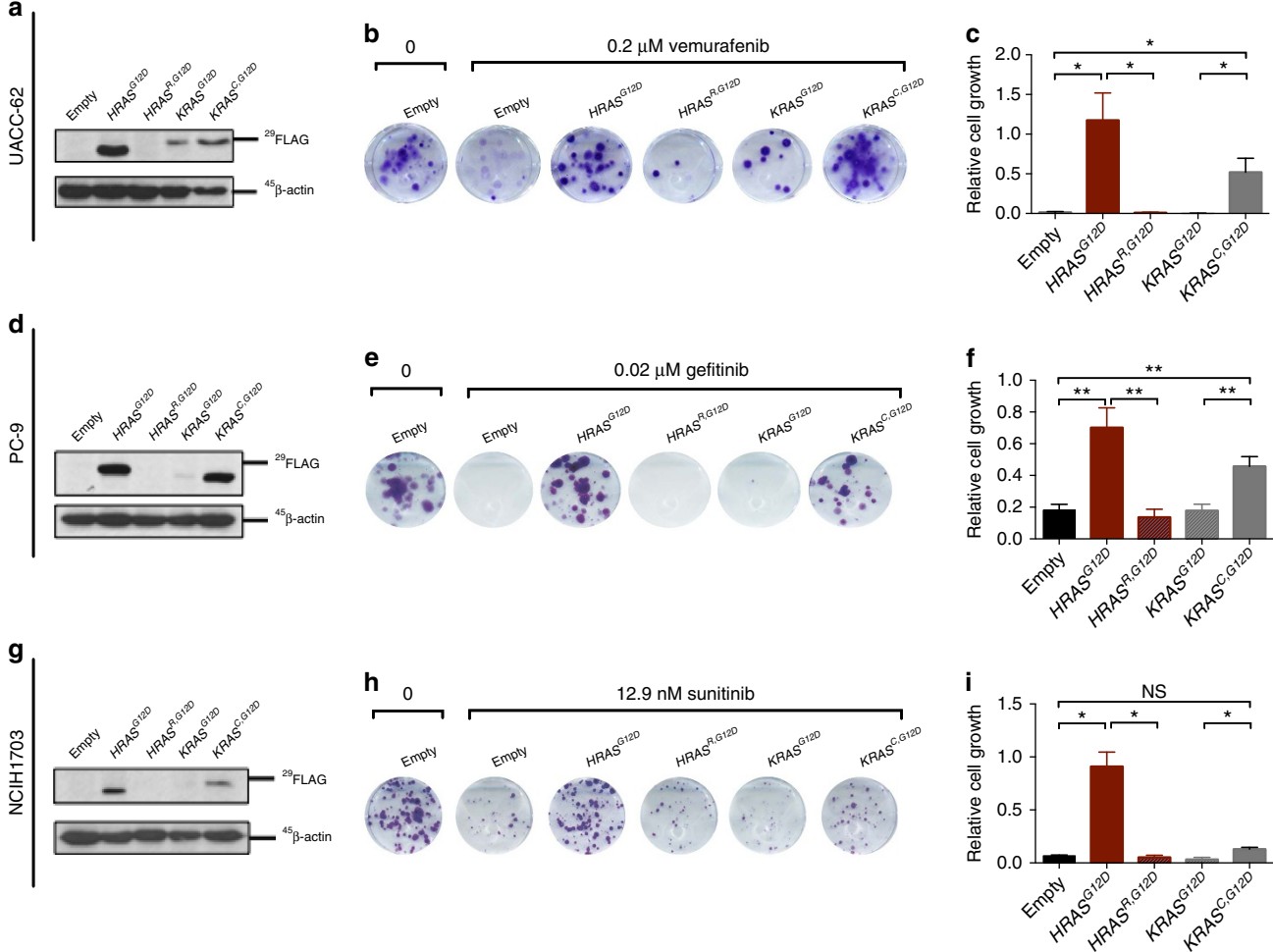

**Figure 2 | Codon bias underlies the differential resistance conferred by oncogenic *HRAS* versus *KRAS*.** (**a,d,g**) Immunoblot analysis with anti-FLAG and anti-β-actin in extracts from indicated cell lines stably transduced with a vector encoding the indicated transgene (or no transgene, empty). All *HRAS* and *KRAS* constructs are FLAG tagged. (**b,e,h**) Representative plate of three technical replicates of colony growth assay of various *HRAS* and *KRAS* expressing derivatives in the presence of indicated drug. (**c,f,i**) Average ± s.e.m. number of colonies in **b,e,h** normalized to *HRAS*[G12D] expressing cells ($n = 3$ technical replicates per condition). Images are cropped for clarity. *$P \leq 0.05$; **$P \leq 0.01$. $P$ values were calculated with unpaired, two-tailed Student's $t$-tests.

effect on MAPK signalling correlated with the degree of cell death, specifically quantification of the extra long-BIM splice variant (EL-BIM) as well as total BIM levels (T-BIM), where decreased induction of this pro-apoptotic protein was observed in common codon-expressing cells when treated with cetuximab, an effect that was lost in the same cells expressing the rare codon counterparts (Fig. 3g, Supplementary Fig. 2D). Furthermore, Annexin V staining revealed that treatment with cetuximab led to significant increases in Annexin V-positive cells in control and rare-codon *RAS*-expressing cells, and that this induction was reduced in cells expressing *HRAS*[G12D] and *KRAS*[C,G12D] (Fig. 3h, Supplementary Fig. 2E). Collectively, these results demonstrate that codon bias underlies the isoform-specific effects of RAS-driven resistance to cetuximab in CRC cell lines, and that common codon-expressing *RAS* isoforms drive resistance, at least partially through the suppression of drug-induced apoptosis. This finding suggests that overcoming poor translation of oncogenic *KRAS* may promote greater resistance to EGFR inhibitors.

**Clinical model reveals selection of potent *KRAS* mutations.** The demonstration that codon bias limits *KRAS*-driven therapeutic resistance in CRC suggests several testable, clinically relevant hypotheses. For example, low levels of KRAS expression may

select for more potent activating mutations that drive acquired resistance without fully overcoming suppressed protein synthesis. Recent work has established that although mutations in *KRAS* codons 12, 13 and 61 are all oncogenic, codon 61 mutations directly disrupt GTPase activity and more potently activate KRAS than allosteric codon 12 and 13 activating mutations[24,25]. Consistent with this idea, although it has been well established that activating mutations in codons 12, 13 and 61 of *KRAS* are all considered drivers of acquired resistance to anti-EGFR mAbs in metastatic CRC patients[17,20,26], there appears to be an unexplained preponderance of codon 61 mutations in patients with anti-EGFR refractory disease, mutations which are seldom observed in patients prior to the initiation of therapy[21,27]. We therefore hypothesized that this detection of Q61 site mutations in tumours with acquired resistance may reflect a selection for more potent mutations to overcome the inherent poor translation of *KRAS*. Indeed, by analysing three separate cohorts of metastatic CRC patients with acquired resistance to anti-EGFR therapy[27–30], we found that in all three groups, and in one combined subset of the groups, there was a significant enrichment in the number of *KRAS*[Q61] site mutations in comparison to a treatment naive background population (Table 1). We next modelled two different point mutations in

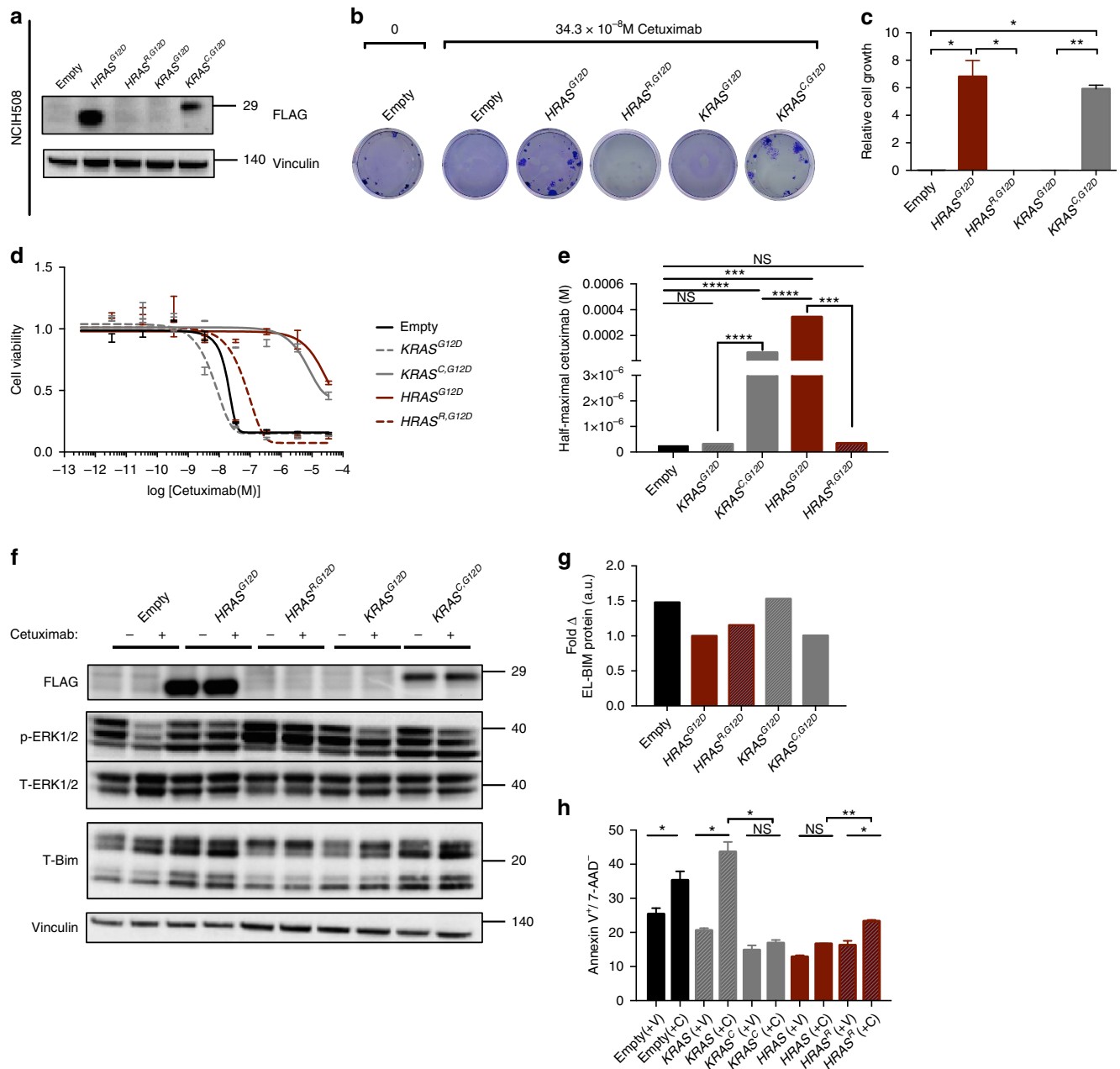

**Figure 3 | Rare codons limit *KRAS*-mediated resistance to cetuximab in colorectal cancer.** (**a**) Immunoblot analysis with anti-FLAG or anti-vinculin antibody in NCIH508 cells stably transduced with a vector encoding the indicated transgene (or no transgene, empty). All *HRAS* and *KRAS* constructs are FLAG tagged. (**b**) Representative plate of three technical replicates of colony growth assay of various *HRAS* and *KRAS* expressing derivatives in the presence of cetuximab or vehicle control. (**c**) Number of colonies in **b** normalized to cells expressing no transgene, determined when the indicated cell line transduced with vectors encoding the indicated transgenes were treated with cetuximab. (**d**) Nonlinear regression growth curves depicting cell viability as detected by Cell Titer Glo of cells transduced with indicated constructs and treated with an eight-log dilution series of cetuximab. (**e**) Derived half-maximal growth inhibition values of curves in **d**. (**f**) Immunoblotting of indicated constructs for signalling of phospho- and total-ERK1/2 and T-BIM. All *HRAS* and *KRAS* constructs are FLAG tagged as in **a**, and vinculin is a loading control. (**g**) Fold change in extra long-BIM (EL-BIM) protein levels from the immunoblots in **f** comparing cetuximab and vehicle treated cells. (**h**) Apoptosis quantification (Annexin $V^+$/7-AAD$^-$) of indicated constructs in the presence or absence of cetuximab. In all cases, data are average ± s.e.m. ($n = 3$ technical replicates per condition). Immunoblotting images are cropped for clarity. *$P \leq 0.05$; **$P \leq 0.01$; ***$P \leq 0.001$, ****$P \leq 0.0001$. $P$ values were calculated with unpaired, two-tailed Student's $t$-tests.

codons 12 or 61 of *KRAS* in the aforementioned *KRAS*^WT^ CRC cell lines (NCIH508 and LIM1215), then calculated the sensitivities of these lines to cetuximab. *KRAS*^Q61^ site mutations were stronger drivers of resistance to anti-EGFR therapy than oncogenic *KRAS*^G12^ site mutations (Fig. 4a,b, Supplementary Fig. 3A,B). Similar to our findings with rare and common

codon-enriched *RAS* constructs, we observed on-target activity of cetuximab in these cells, as evidenced by reduced p-ERK levels (Fig. 4c, Supplementary Fig. 3C) and induction of EL-BIM and T-BIM that was suppressed by the resistance-conferring *KRAS*^Q61^ mutations (Fig. 4d, Supplementary Fig. 3D). Importantly, these effects were independent of KRAS protein expression levels,

**Table 1 | KRAS mutational status prevalence among treatment naive and treatment resistant cohorts.**

|  | KRAS mutation | No. (patients) | Total | P value |
|---|---|---|---|---|
| Treatment naive[28] | G12/13 | 76 | 224 |  |
|  | **Q61** | **4** | **224** |  |
| Treatment resistant[29] | G12/13 | 20 | 24 | 0.0108 |
|  | **Q61** | **10** | **24** | **<0.0001** |
| Treatment resistant[30] | G12/13 | 3 | 16 | 0.42 |
|  | **Q61** | **8** | **16** | **<0.0001** |
| Treatment resistant[27] | G12/13 | NA | NA | **<0.0001** |
|  | **Q61** | **9** | **27** |  |
| Treatment resistant[29,30] | G12/13 | 23 | 40 | 0.0864 |
|  | **Q61** | **18** | **40** | **<0.0001** |

P values as assessed by two-tailed Fisher's test. NA, not applicable. Superscripts signify reference number.

which were similarly low across the various 12 and 61 point mutations tested (Fig. 4e, Supplementary Fig. 3E). Together, these data suggest that selection for stronger KRAS mutations can overcome poor KRAS expression imposed by codon bias, a result that may explain the enrichment of $KRAS^{Q61}$ site mutations in patients with acquired cetuximab resistance.

**KRAS expression determines intrinsic resistance to cetuximab.** Aside from the possibility of selection for a more potent oncogenic mutation, another potential mechanism to overcome the inability of oncogenic KRAS to potently impart drug resistance is by producing more of the protein. As such, primary KRAS-driven resistance to cetuximab may be dependent not only on the presence of activating mutations, as is commonly understood[17], but also on the degree to which cells surmount poor KRAS expression prior to therapy. To test this hypothesis, we examined a panel of 21 $KRAS^{G12/G13}$ mutant CRC cell lines that were previously characterized for their intrinsic sensitivity to cetuximab[31] to determine whether the degree of resistance to this drug correlated with the expression level of mutant KRAS protein. Indeed, after stratifying cell lines on the basis of responsiveness to cell growth inhibition in the presence of cetuximab (with responsive lines being defined as exhibiting >10% growth inhibition by cetuximab and non-responsive lines showing no growth inhibition), we found a significant elevation in KRAS expression in non-responsive cell lines relative to responsive lines (Fig. 5a,b). We then compared global translation rates in cetuximab responsive and non-responsive cell lines by labelling with the translation elongation inhibitor puromycin, which leads to direct puromycin incorporation into nascent polypeptides that can be detected with anti-puromycin antibodies[32]. Following 15 min of puromycin treatment, immunoblotting revealed the rate of global translation to be higher in cetuximab non-responsive cell lines than cetuximab responsive cell lines, and quantification of labelling showed a significant correlation with KRAS expression in these cell lines (Supplementary Fig. 4A,B). This observed correlation between intrinsic resistance, KRAS protein expression and global protein synthesis suggests that KRAS mutational status alone may not fully predict primary resistance to EGFR inhibition in CRC. Instead, the expression level of mutant KRAS is likely to be a second key contributor to resistance, a notion that directly implicates codon bias and the ability of cancer cells to adopt mechanisms that overcome the translational barrier it imposes.

To begin discerning what cellular programs might be involved in upregulating mutant KRAS protein expression, we analysed a broader panel of KRAS-mutant lung, pancreatic and CRC cell lines, and observed that lines that displayed high endogenous KRAS protein expression (KRAS-high lines) also tended to exhibit elevated expression of an ectopic KRAS transgene when compared to cell lines with low endogenous KRAS protein expression (KRAS-low lines) (Fig. 5c). This was unexpected, as KRAS cDNAs are typically expressed poorly in the absence of codon optimization (refer to Fig. 1), suggesting the intriguing possibility of a mechanism(s) in these cells to overcome poor KRAS expression that is transferable to a plasmid-borne KRAS cDNA. To investigate whether this feature reflects a more general strategy by cancer cells dependent on mutant KRAS expression to overcome poor oncogene translation, we next stably expressed an AKT3 cDNA, which is similarly enriched in rare codons and poorly expressed[22], in this panel of KRAS mutant cell lines. Interestingly, many KRAS-high lines also displayed elevated levels of AKT3 when compared to their KRAS-low counterparts (Fig. 5d, Supplementary Fig. 4C). By contrast, we did not observe a correlation between expression levels of ectopic KRAS and ectopic NRAS, the latter of which contains approximately equal percentages of rare and common codons (Supplementary Fig. 4D,E). To further test this concept, we stably expressed the gene pair ORMDL3/1, which is analogous to HRAS/KRAS in that it exhibits divergent codon bias and protein expression[22], in KRAS-high/low cells. Divergent codon-dependent expression of ORMDL1/3 was completely normalized in the KRAS-high cell line SW900, even in the absence of codon optimization of the rare codon-enriched ORMDL1, but was not observed in the KRAS-low cell line AsPC1 (Fig. 5e). Finally, to compare global translation in these KRAS-high/low lines, a time course of puromycin labelling was used, revealing the rate of global translation to be higher in KRAS-high cells (SW900) than KRAS-low cell cells (AsPC1) (Fig. 5f). Collectively, these data suggest that cancer cells can overcome poor KRAS expression by globally upregulating protein synthesis through mechanisms that also enable cancer cells to overcome the effects of codon bias.

**Selective targeting of anti-EGFR resistant colorectal cancer.** The above results suggest the intriguing possibility that tumours with acquired $KRAS^{G12}$ or $KRAS^{G13}$ site mutations must overcome poor translation to achieve resistance. To test this hypothesis, we measured the levels of KRAS protein in the parental LIM1215 cell line and in two independently derived, $KRAS^{G12/G13}$ mutant clones selected for resistance to cetuximab[13]. In both clones, KRAS protein levels were higher than the parental cells (Fig. 6a–c). The same result was observed when the experiment was repeated using the cell line OXCO2 (Supplementary Fig. 5A,B). Consistent with our previous observations, upregulated KRAS expression in resistant derivatives was associated with increased global translation rate (Supplementary Fig. 5C). Upregulation of global translation is sufficient to increase KRAS protein expression levels, as ectopic expression of eIF4E, a key factor in protein synthesis, drove increases in KRAS expression in parental LIM1215 cells to levels comparable to those observed in a resistant derivative (Supplementary Fig. 5D). (We note that, as expected, eIF4E-driven upregulation of $KRAS^{WT}$ expression was insufficient to drive resistance to cetuximab, as resistance also requires mutational activation of KRAS (Supplementary Fig. 5E).) Given the evidence that cells overcome KRAS codon bias by upregulating translation (Fig. 5), we hypothesized that inhibitors of translation may be selectively potent in CRC cells with acquired $KRAS^{G12}$ or $KRAS^{G13}$ site mutations, thereby revealing a synthetic lethality of the resistant state and a potential target in anti-EGFR resistant disease. We selected a

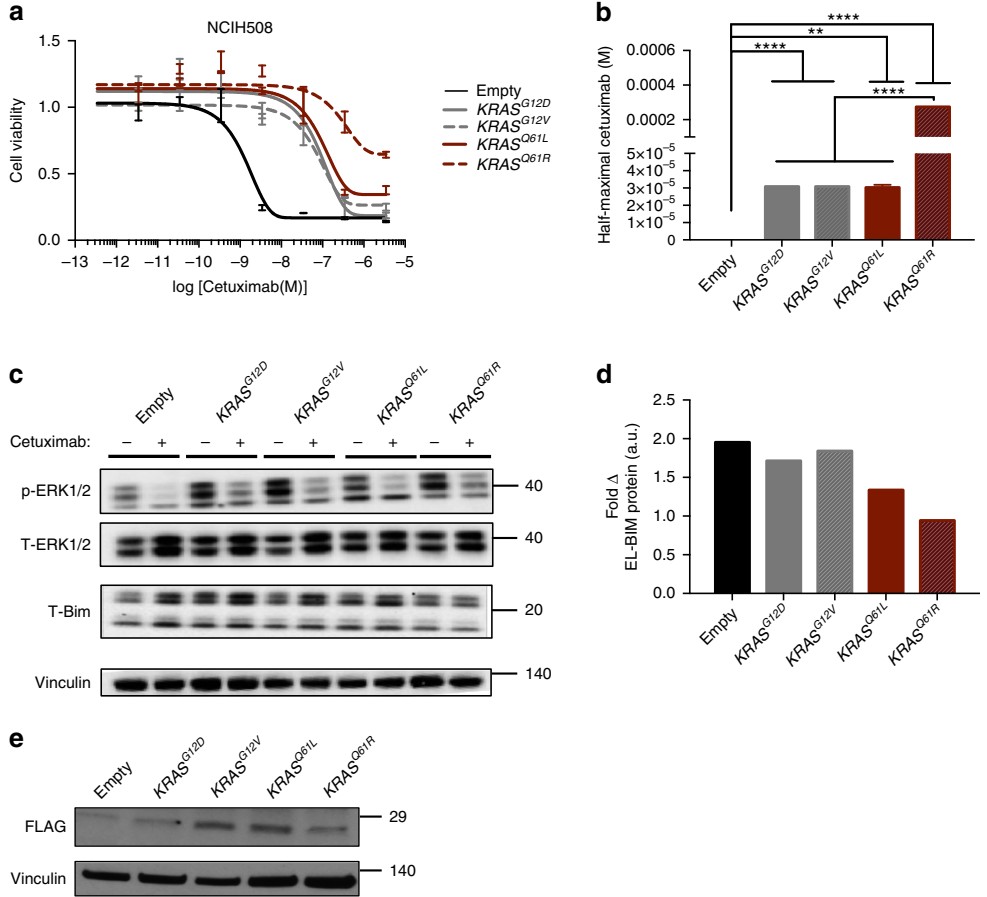

**Figure 4 | Model of clinical *KRAS*-mediated resistance to cetuximab reveals selection of more potent *KRAS* mutations.** (**a**) Nonlinear regression growth curves depicting cell viability as detected by Cell Titer Glo of NCIH508 cells transduced with a vector encoding no transgene (empty) or the indicated RAS transgene in the presence of cetuximab. (**b**) Derived half-maximal growth inhibition values of curves in **a**. (**c**) Immunoblotting of indicated constructs for phospho- and total-ERK1/2 and T-BIM in NCIH508 cells. (**d**) Fold change in extra long BIM (EL-BIM) protein levels from the immunoblots in (**c**) comparing cetuximab and vehicle treated cells. (**e**) Immunoblotting of indicated constructs for FLAG-tagged KRAS. In all cases, data are average ± s.e.m. ($n = 3$ technical replicates per condition). Images are cropped for clarity. **$P \leq 0.01$; ****$P \leq 0.0001$. $P$ values were calculated with unpaired, two-tailed Student's $t$-tests.

panel of translational inhibitors, including Rapamycin (an allosteric mTORC1 inhibitor), AZD2014 and MLN0128 (ATP-competitive mTORC1/2 inhibitors), BEZ235 (a dual ATP-competitive mTOR/PI3K inhibitor), and 4EGI-1 (a direct eIF4E inhibitor), and found that in all cases, acquired-resistant derivatives were significantly more sensitive to these inhibitors. For the case of allosteric and ATP-competitive mTOR inhibitors in particular, many of which are FDA approved or in advanced clinical development, we observed growth inhibition of resistant clones at doses 10- to 100-fold lower than their matched parental counterparts (Fig. 6d). Further, levels of KRAS protein were diminished in resistant cells when compared to parental cells within 4 h of treatment with each translational inhibitor (Fig. 6e). KRAS is differentially required for the growth of resistant cells relative to their parental counterparts as evidenced by short hairpin RNA (shRNA) knockdown experiments (Supplementary Fig. 5F,G). As such, the enhanced growth inhibitory effects of translational inhibitors in resistant cells parallels the effects of KRAS knockdown. Together, these data suggest that in CRC cells with *KRAS*$^{G12/G13}$ mutation-driven acquired resistance, a translationally primed state overcomes codon bias and facilitates expression of mutant KRAS at levels sufficient to potentiate resistance. Thus, the targeting of this translation-dependent state may provide a novel, selective therapeutic target for the treatment of anti-EGFR refractory disease.

## Discussion

Despite providing a clinically significant survival benefit to patients with metastatic *KRAS* WT CRC, acquired resistance to the anti-EGFR mAbs cetuximab and panitumumab can arise via three predominant mechanisms: (1) genetic alterations involving downstream EGFR effectors, including *KRAS*, (2) activation of parallel receptor tyrosine kinase pathways, including HER2 and c-MET and (3) mutations in the extracellular domain of *EGFR*[33]. *KRAS* mutations appear to be the most common driver of acquired resistance to anti-EGFR mAbs. In one study, serial sampling of patient sera revealed that 9 out of 24 (38%) patients with initially *KRAS* WT tumours developed detectable mutations in *KRAS* between 5 and 6 months after the initiation of panitumumab monotherapy[34]. In another study, analysis of metastases from patients who developed resistance to anti-EGFR mAbs showed the emergence of *KRAS* amplification in one sample and acquisition of secondary *KRAS* mutations in 60% (6 out of 10) of the cases examined[13]. Although *KRAS* has been shown to play a central role in primary and acquired resistance to anti-EGFR therapy in CRC as well as in other therapeutic contexts, no effective pharmacological inhibitors of KRAS, or other RAS oncoproteins, have reached the

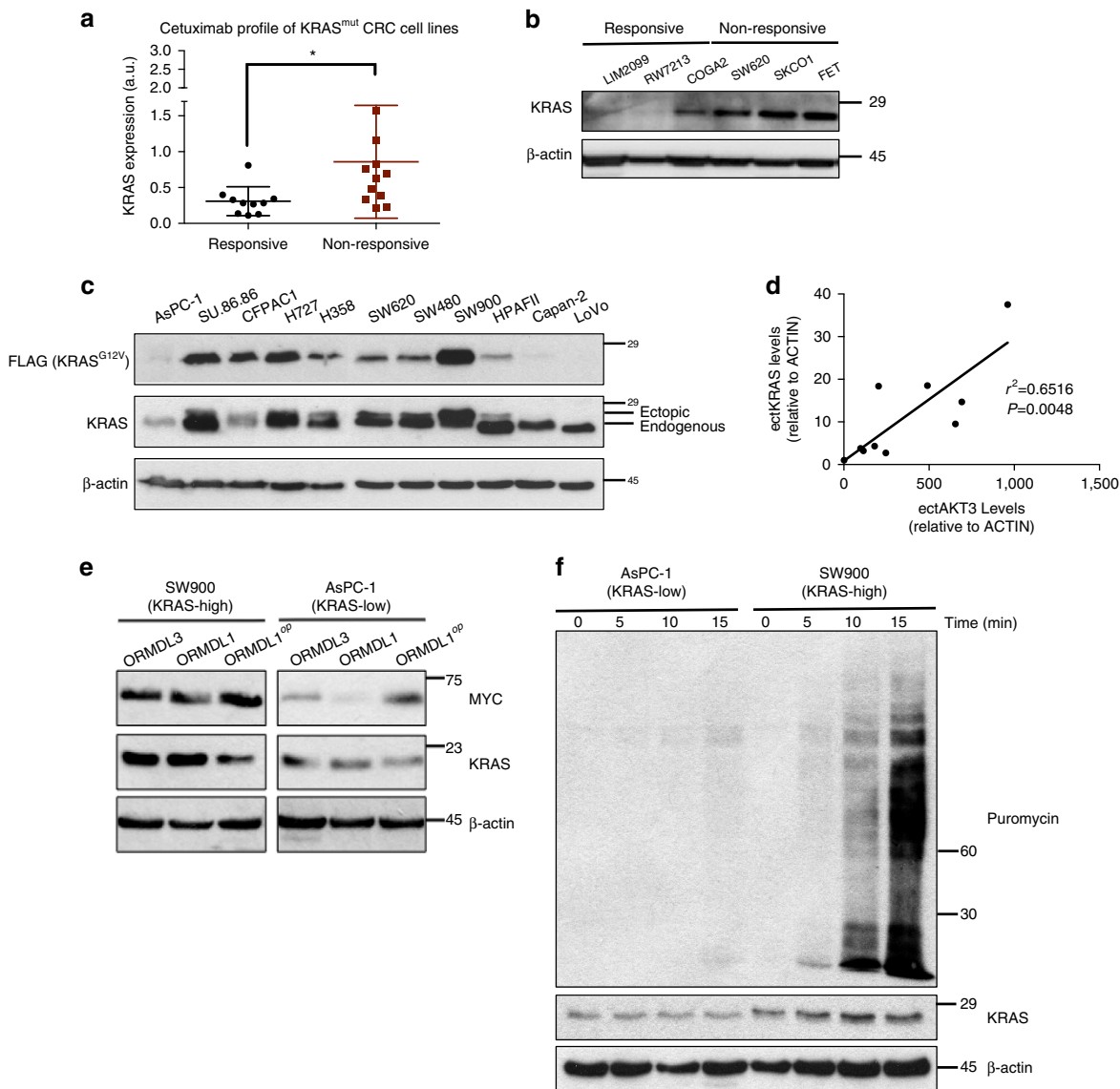

**Figure 5 | Expression levels of mutant *KRAS* correlate with intrinsic resistance to cetuximab and are mediated by translational upregulation.**
(**a**) Endogenous KRAS protein expression (normalized to β-actin) in 21 *KRAS* mutant CRC cell lines classified as responsive (>10% growth inhibition following treatment with cetuximab, ten cell lines) or non-responsive (no growth inhibition following treatment with cetuximab, 11 cell lines).
(**b**) Immunoblot analysis for endogenous KRAS in representative responsive or non-responsive cell lines. (**c**) Immunblot analysis for endogenous WT *KRAS* and exogenous FLAG-tagged *KRAS*^*G12V* in a panel of *KRAS* mutant cell lines. (**d**) Linear correlation plot of ectopic MYC-AKT3 protein levels and ectopic FLAG-KRAS^*G12V* protein levels in cell lines. Immunoblots corresponding to these data are presented in Supplementary Fig. 4. (**e**) Immunoblotting of indicated MYC-tagged constructs in representative KRAS-high/low lines, where 'op' signifies codon-optimized construct. (**f**) Immunoblotting time course of puromycin labelling in representative KRAS-high/low lines. Images are cropped for clarity. Error bars show data ± s.e.m. ($n = 3$ technical replicates per condition). *$P \leq 0.05$. $P$ values were calculated with unpaired, two-tailed Student's *t*-tests.

clinic to date[35]. This stands in contrast with the other predominant mechanisms of anti-EGFR resistance. For example, alternative receptor tyrosine kinase pathways such as HER2 and c-MET can be targeted with selective TKIs[36]. Similarly, extracellular domain mutations in *EGFR*, which occur in about 20% of CRC patients treated with anti-EGFR mAbs[33], can be treated with oligoclonal antibodies like MM-151 and Sym004, which have been shown to overcome this resistance mechanism in model systems and are now being explored clinically as secondary therapies for patients who have relapsed on anti-EGFR mAbs[33,37,38]. Thus, there is a particularly compelling need for therapeutic strategies that block *KRAS*-driven resistance to EGFR inhibitors in CRC.

Our findings suggest that the resistance-conferring ability of mutant *KRAS* is limited by codon bias, a mechanism that controls the expression and subsequent downstream activities of RAS proteins. Indeed, cells with acquired cetuximab resistance driven by canonical *KRAS*^*G12/G13* site mutations exhibit higher levels of this protein. Alternatively, the selection for more potent *KRAS* mutations is an alternative mechanism to overcome codon bias, a concept that may explain the paradoxical enrichment of *KRAS*^*Q61* mutations observed in patients with anti-EGFR refractory disease. Combined, these studies suggest that *KRAS*-mediated therapeutic resistance may require higher levels of KRAS activity than *KRAS*-mediated tumorigenesis, an idea that is consistent with other signal amplification-based resistance mechanisms[39]. To examine these concepts further, future studies to determine the direct relationship between KRAS expression levels in patient tumours and subsequent clinical response to anti-EGFR mAbs are

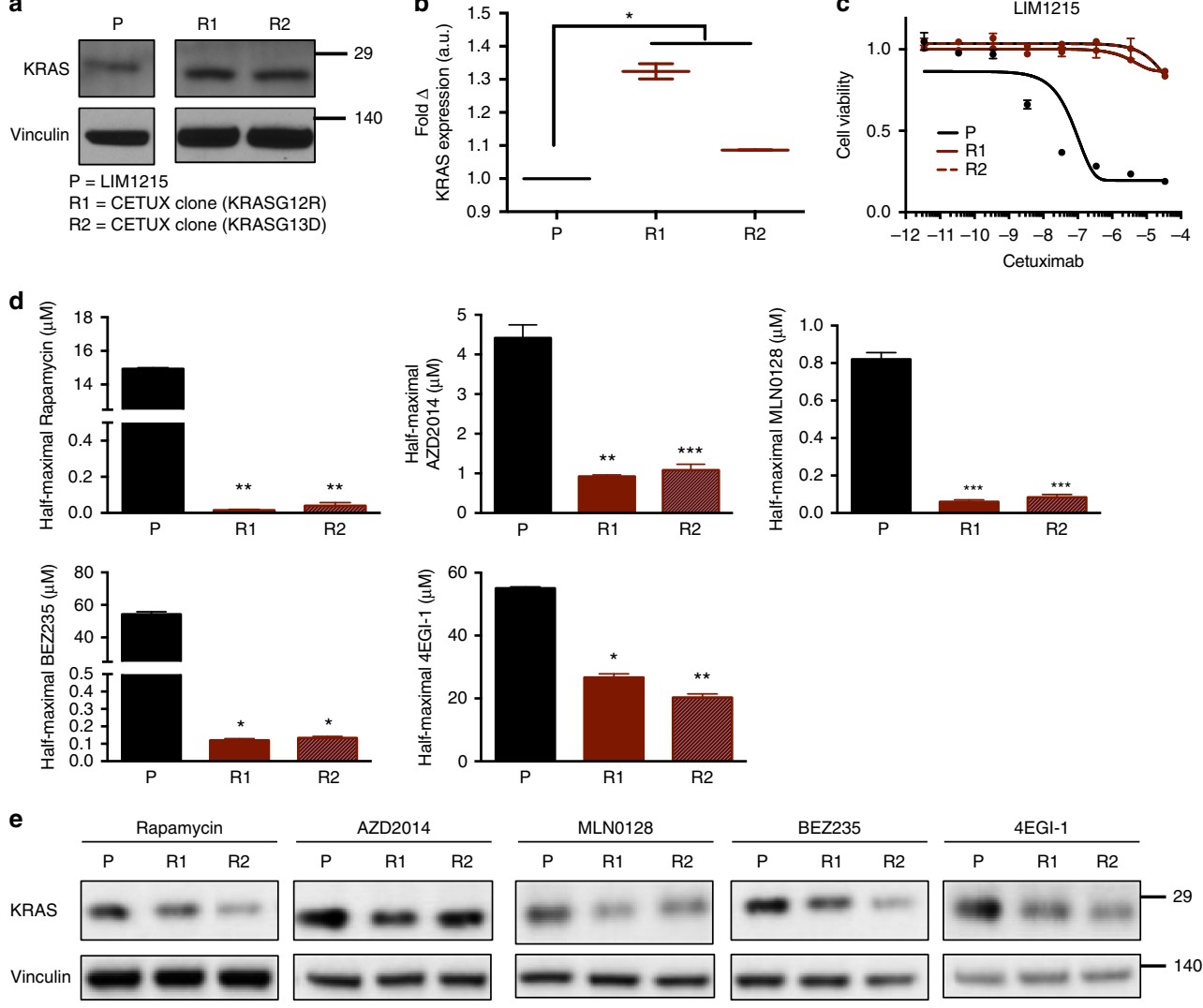

**Figure 6 | Translation-dependent state of *KRAS*$^{G12/G13}$-mutant cells enables selective targeting of anti-EGFR resistant colorectal cancer.**
(**a**) Immunoblot analysis for endogenous KRAS levels in parental (P) and derived *KRAS* mutant, cetuximab-resistant (R1 and R2) clones. (**b**) Fold change in KRAS protein levels, normalized to loading control, between resistant derivatives (R1 and R2) and parental cells (P). (**c**) Nonlinear regression growth curves depicting cell viability as assessed by Cell Titer Glo (CTG) of parental (P) and matched resistant derivatives (R1 and R2) treated with cetuximab. (**d**) Derived half-maximal growth inhibition values of parental (P) and matched resistant derivatives (R1 and R2) treated with Rapamycin, AZD2014, MLN0128, BEZ235 and 4EGI-1. (**e**) Immunoblot analysis of KRAS protein levels in parental (P) and resistant derivatives (R1 and R2) following 4 h treatment with the corresponding inhibitors in **d**. Images are cropped for clarity from the same exposure of the same membrane. Error bars show data ± s.e.m. ($n = 3$ technical replicates per condition). *$P \leq 0.05$; **$P \leq 0.01$; ***$P \leq 0.001$. P values were calculated with unpaired, two-tailed Student's $t$-tests.

warranted. Similarly, direct comparison of KRAS expression levels before and following the onset of *KRAS*$^{G12}$-mediated acquired resistance will shed further light on the universality of translational upregulation in this context.

Previously, it has been demonstrated that oncogenic activation of translation initiation and/or elongation can support tumorigenesis by driving selective upregulation of specific mRNA transcripts[40]. Indeed, seminal studies revealed that overexpression of eIF4E, involved in the translation initiation complex of protein synthesis, was sufficient to drive tumorigenesis in cell lines and spontaneous tumorigenesis in mice[41–43]. In addition, deregulation of translation has more recently been identified as both a primary downstream consequence of oncogenic signalling, as well as a central mediator of resistance to clinical therapies, including drugs targeting the MAPK and PI(3)K-AKT-mTOR signal transduction

pathways[44,45]. These findings, along with data showing that cancer cells co-opt translational machinery to support tumour growth, have driven increased interest in targeting translational control as a method to selectively kill cancer cells[46].

By treating cells with direct and/or indirect inhibitors of protein synthesis, it is possible to selectively target the translationally primed state that facilitates *KRAS*$^{G12/G13}$-mediated acquired resistance. Although our studies indicate that these resistant cells exhibit hypersensitivity to diverse inhibitors of translation, future studies are required to determine the reason for this sensitivity and the specific mechanisms imparting greater translation of KRAS, which may lead to more precise and refined approaches to target these tumours. Importantly, however, indirect inhibitors of translation such as rapamycin and MLN0128, and their analogues, are already in clinical development. As such, studies to examine the sensitivity of

**Table 2 | TRC shRNA sequences.**

| Construct | TRC ID | Sequence |
|---|---|---|
| 1. shKRAS-1 | TRCN0000033262 | CCTATGGTCCTAGTAGGAAAT |
| 2. shKRAS-2 | TRCN0000040151 | CCTACAGGAAGCAAGTAGTAA |

patients with $KRAS^{G12/G13}$-mediated resistance to these agents, or the ability of upfront treatment with these agents in combination with anti-EGFR mAbs to shape resistance evolution, are warranted.

## Methods

**Cell lines and reagents.** All cell lines were grown at 37 °C in 5% $CO_2$. A375, PC9, NCIH508, UACC-62, H1073, LIM1215, SW620, SW480, SW900, HPAFII, Capan-2, AsPC-1, SU.86.86, CFPAC1, H727, H358, OXCO2 and LoVo were cultured in RPMI supplemented with 10% FBS and 1% penicillin/streptomycin. SKBR3 was cultured in McCoy's 5a supplemented with 10% FBS and 1% penicillin/streptomycin. All other cell lines were purchased from American Type Culture Collection (ATCC) or Duke University Cell Culture Facility (CCF). LIM1215, OXCO2 and matched anti-EGFR resistant derivatives, as well as KRAS mutant CRC lines and cetuximab, were used as previously described[18]. All other drugs were purchased from Selleck Chemicals, ChemieTek, MedChemExpress, Ontario Chemicals, Sigma-Aldrich, or Apex Bio.

**Colony formation and soft agar growth assays.** For colony formation assays, UACC-62, PC-9, H1073 and NCI-H508 cells selected for stable expression of the indicated RAS constructs were seeded in duplicate into six-well plates at 250 cells per well. Twenty-four hours later, DMSO or the indicated drug (in DMSO) were added to cells. DMSO or drug-containing media was replaced every 2 days and the assays were cultured for 14–21 days. Plates were then rinsed with phosphate-buffered saline (PBS), fixed in 10% formalin for 5 min and stained with 0.1% crystal violet stain for 30 min. Plates were rinsed in distilled water and scanned. ImageJ software was used to quantify colony growth area as a percentage of the well covered. Anchorage-independent growth was assayed in six-well plates with 1 ml of 0.6% bactoagar media solution (final concentration $1 \times$ DMEM, 10% FBS, $1 \times$ penicillin/streptomycin) as a bottom support layer. A total of $4 \times 10^4$ UACC-62 cells selected for stable expression of indicated RAS constructs were resuspended in RPMI (10% FBS, $1 \times$ penicillin/streptomycin) and mixed 1:1 with 0.6% bactoagar media solution to give a final bactoagar concentration of 0.3%, 0.2 µM vemurafenib (or equivalent amount DMSO). Cells were then plated in triplicate to give final density of $1 \times 10^4$ cells per well. Each well was fed with fresh media and drug on days 7 and 14. Colonies were counted on day 21.

**Short-term growth-inhibition assay (GI50).** Cells were seeded into 96-well plates at 1,000 cells per well for inhibition assays to cetuximab or at 5,000 cells per well for all other drugs tested. To generate $GI_{50}$ curves, cells were treated with vehicle (PBS or DMSO) or an eight-log serial dilution of drug. Each treatment condition was represented by at least three replicates. Seven days after cetuximab addition or 3 days after all other drug additions, cell viability was measured using Cell Titer Glo (Promega). Relative viability was then calculated by normalizing luminescence values for each treatment condition to control treated wells. To generate $GI_{50}$ curves for exogenously expressed KRAS experiments, slight modifications were made. Cells were seeded at 300,000 cells per well in six-well plates. The following day cells were infected with the desired retroviral constructs. Following 2 days of puromycin selection ($2 \mu g ml^{-1}$), the cells were seeded into 96-well plates at 5,000 cells per well. Dose–response curves were fit using GraphPad/Prism 6 software.

**Western blotting and antibodies.** Immunoblotting was performed as previously described[47], and membranes were probed with the following primary antibodies at the corresponding dilutions: vinculin, β-actin, BIM, p-ERK1/2, ERK, p-AKT (S473,T308), AKT, eIF4E at 1:1,000 (Cell Signaling); HRAS, NRAS and KRAS at 1:100 (Santa Cruz Biotechnology); FLAG at 1:1,000 (M2; Sigma Aldrich); Myc at 1:1,000 (Invitrogen) and Puromycin at 1:10,000 (Millipore). To determine correlation between ectKRAS and ectAKT3 protein levels, immunoblots were scanned and quantified using ImageJ software. For each cell line analysed, the band density of ectKRAS and ectAKT3 (relative to actin) was determined and plotted graphically. For quantification of EL-BIM, T-BIM and KRAS protein levels, immunoblots were scanned, quantified using ImageJ software, and the band density of each protein (relative to vinculin) was determined and plotted graphically.

**Pathway activating screens.** Data are excerpted from screens previously reported[12,18,23].

**Puromycin incorporation.** Cell lines were cultured to ~70% confluence were treated with DMSO or 5 µM puromycin for the indicated time periods. After incubation, cells were washed with PBS to halt puromycin incorporation. Cells were lysed and subjected to immunoblotting with anti-puromycin antibody (Millipore, 1:10,000).

**Lentiviral and retroviral production and infection of cells.** Experiments performed using HRAS, KRAS and NRAS point mutants and codon-modified constructs were conducted as previously described[22,48]. Lentiviral experiments were performed as previously described[23,49]. In brief, HEK293T cells were transfected with a mixture of VSVG, PsPAX2, the construct of interest, and Opti-MEM and FuGENE 6 Transfection Reagent. The following day, transfection media was replaced with 20 ml of harvest media (RPMI + 30% FBS + 1% PS) collected at 48 h post media change. Collected virus was passed through a 0.45 µM filter and frozen at − 80 °C prior to use in infection of various cell lines. We note here that the ability to successfully ectopically express oncogenic RAS was variable from cell line to cell line.

**shRNA constructs.** TRC shRNA clones (Table 2) were obtained from the Duke RNAi Facility as glycerol stocks. Constructs were prepared in lentiviral form and used to infect target cells, as previously described[50].

**Quantification of apoptosis by Annexin-V.** Cells were seeded in 12-well plates and treated the next day with no agent, cetuximab ($34.3 \times 10^{-8}$ M) or vehicle (PBS). Cells were incubated for 5 days and then washed twice with ice-cold PBS, and resuspended in $1 \times$ Annexin V binding buffer (10 mM HEPES, 140 mM NaCl, 2.5 mM $CaCl_2$; BD Biosciences). Surface exposure of phosphatidylserine was measured using APC-conjugated Annexin V (BD Biosciences). 7-AAD (BD Biosciences) was used as a viability probe. Experiments were analysed at 20,000 counts per sample using BD FACSVantage SE. Gatings were defined using untreated/unstained cells as appropriate.

**Statistics.** Results are expressed as the means ± s.e.m. Unless otherwise specified, for comparisons between two groups, P values were calculated with unpaired, two-tailed Student's t-tests.

**Data availability.** All data generated or analysed during this study are included in this published article (and its Supplementary Information files).

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

## Acknowledgements

We thank Channing Der, Adrienne Cox, Carlotta Cancelliere, and members of the Bardelli, Counter, and Wood labs for helpful discussions and technical assistance. This work was supported by Duke University School of Medicine start-up funds (K.C.W.), a scholar award from the NIH Building Interdisciplinary Research Careers in Women's Health Program (K12HD043446, K.C.W.), a Golfers Against Cancer Research Award (K.C.W.), a Stewart Trust Fellowship (K.C.W.), and research grants from the Pancreatic Cancer Action Network-American Association for Cancer Research (K.C.W.), NIH (U01CA199235 and R01CA207083 to K.C.W., R01CA12031, R01CA154630 and P01CA203657 to C.M.C., and F32CA192715 to E.K.), NSF (DGE-1106401, G.R.A.), the Duke Clinical and Translational Science Award (UL1TR001117, K.C.W.), AIRC 2010 Special Program Molecular Clinical Oncology 5 per mille, Project no. 9970 (A.B.), Fondazione Piemontese per la Ricerca sul Cancro-ONLUS 5 per mille 2011 Ministero della Salute (A.B.), and AIRC IG n. 16788 (A.B.).

## Author contributions

Conceptualization: M.A., E.K., C.M.C. and K.C.W.; experimentation: M.A., E.K., G.R.A. and S.J.S.; formal analysis: all authors; writing of original manuscript: M.A., C.M.C. and K.C.W.; writing, review and editing: all authors; resources: S.A. and A.B.; supervision: A.B., C.M.C. and K.C.W. M.A. and E.K. contributed equally as co-first authors. G.R.A. and S.J.S. contributed equally as co-second authors.

## Additional information

**Competing interests:** The authors declare no competing financial interests.

