## [Peer Review File (PDF 456 kb) · Nature Communications]

Reviewers' comments:

Reviewer #1 (Remarks to the Author):

Although this work is an extension of previous reports by these authors, in my opinion, the authors report important and translationally relevant implications of this biology that will be of interest to the oncology field in general and signal transduction researchers in particular. The observations appear to be robust and have been effectively described. I have no major concerns.

Reviewer #2 (Remarks to the Author):

In this manuscript, Ali et al. describe a role for codon usage in restraining KRAS expression as a barrier to drug resistance. The authors employ a pharmacological approach to show that overexpression of HRAS can drive drug resistance, while the closely related KRAS gene is less potent in driving drug resistance. The authors then show that expression of codon-optimized KRAS leads to higher protein levels, and confers drug resistance, similar to the effect observed when HRAS is overexpressed, implicating the codon usage of KRAS as a determinant of protein levels and drug resistance. Furthermore resistant cell lines with weakly activating KRASG12/13 mutations tend to have increased KRAS expression, and increased global protein synthesis rates. This is in contrast to the effect of strongly activating mutations such as KRASQ61R, which can confer drug resistance despite being expressed at low levels. Interestingly the authors note that cetuximab-resistant cell lines with weakly activating G12/13 mutations, as compared to the non-resistant parental line, exhibit increased KRAS expression. Importantly, cetuximab resistant cell lines are more sensitive than the parental lines to treatment with various inhibitors of protein synthesis. This effect is associated with a reduction in KRAS levels, suggesting that the drugs may kill at least in part through reducing KRAS expression.

Overall, the link between codon usage and RAS expression is established through well-designed experiments in multiple cell lines, suggesting a general role for codon usage in regulating RAS expression. The observation that strongly-activating KRASQ61R mutations can drive resistance, despite low expression, while weakly-activating KRASG12/13 mutations require increased expression levels is very interesting, and suggests a novel mechanism of RAS-dependent drug resistance. However, the link between "translational upregulation" and increased KRAS levels is correlative. The manuscript would be greatly strengthened if the authors would provide evidence showing that the increased KRAS levels are the result of augmented protein synthesis, and are sufficient to confer drug resistance. Additionally, it is not clear if the lethality of drugs targeting the translational apparatus is due specifically to the decrease in KRAS levels, or due to either an overall decrease in protein synthesis or other targets affected by the compounds employed in the experiments. In principle, the manuscript is suitable for publication in Nature Communications, but these points should be addressed prior to acceptance.

Major comments:

1. In Figure 5, the authors show a correlation between global protein synthesis rates and KRAS expression, but it is not clear if the relationship between the two is causal. The observation that mTOR inhibitors abrogate KRAS expression implicates increased translation initiation and/or elongation rates in driving KRAS expression. However, mTOR kinase phosphorylates many targets including components of the protein synthesis machinery, therefore a clearer mechanistic understanding of the causal effectors downstream of mTOR would strengthen the manuscript. For example, the authors could use a non-drug resistant cell line (i.e. LIM1215) and overexpress eIF4E and/or eEF2, to see if upregulation of a specific step in translation can increase KRAS levels and drive drug resistance.

2. In Figure 5, the link between global protein synthesis rate, cetuximab resistance, and KRAS

expression is supported by data from only 2 cell lines. The authors should determine global protein synthesis rates in the panel of cell lines tested in Figure 5A, to see if the correlation between drug resistance and KRAS levels is a general trend.

3. In Figure 5, it would be beneficial if the authors could provide an example of a gene not dependent on codon usage, to see if its expression is affected by “translational upregulation”. This would suggest that in the context of this cell line the effect of translational upregulation is specific to genes dependent on codon usage.

4. In Figure 6, it is not clear if treatment with translational inhibitors is killing the cells specifically through downregulation of KRAS, or through the inhibition of other targets. The authors should perform rescue experiments in the resistant cell lines by overexpressing HRAS, codon-optimized KRAS, and wild type KRAS. If the drugs kill through blocking the translational upregulation of KRAS, then expression of a HRAS or codon optimized KRAS, which are efficiently expressed, should rescue the killing effect of the translational inhibitors, whereas a wild type KRAS should not.

5. For Figure 6, it is not clear if “translational upregulation” is even occurring in these specific cell lines. An essential experiment would be to assay global protein synthesis rates in the parental and resistant cell lines.

6. Throughout the manuscript, conclusions based on western blots (e.g. Figs 1E, 2A, D, G, 3F, etc.) are often presented without quantification and statistical analysis. It is not clear if the results are representative of multiple experiments, or if the conclusions are only supported by what appear to be single experiments. Indeed, for any quality publication, multiple independent experiments should be performed in order to substantiate the authors’ claims.

Minor comments:

1. Figure 1D is not interpretable in its current state. The data should be presented in a clearer way.

2. In figure 6D, the authors switch between showing GI75 and GI50 values for different drugs. One or both values should be shown for all drug treatments.

Reviewer #1 (Remarks to the Author):

Although this work is an extension of previous reports by these authors, in my opinion, the authors report important and translationally relevant implications of this biology that will be of interest to the oncology field in general and signal transduction researchers in particular. The observations appear to be robust and have been effectively described. I have no major concerns.

We thank Reviewer #1 for his/her review of our manuscript and appreciate his/her interest in the significance of our studies.

Reviewer #2 (Remarks to the Author):

In this manuscript, Ali et al. describe a role for codon usage in restraining KRAS expression as a barrier to drug resistance. The authors employ a pharmacological approach to show that overexpression of HRAS can drive drug resistance, while the closely related KRAS gene is less potent in driving drug resistance. The authors then show that expression of codon-optimized KRAS leads to higher protein levels, and confers drug resistance, similar to the effect observed when HRAS is overexpressed, implicating the codon usage of KRAS as a determinant of protein levels and drug resistance. Furthermore resistant cell lines with weakly activating KRASG12/13 mutations tend to have increased KRAS expression, and increased global protein synthesis rates. This is in contrast to the effect of strongly activating mutations such as KRASQ61R, which can confer drug resistance despite being expressed at low levels. Interestingly the authors note that cetuximab-resistant cell lines with weakly activating G12/13 mutations, as compared to the non-resistant parental line, exhibit increased KRAS expression. Importantly, cetuximab resistant cell lines are more sensitive than the parental lines to treatment with various inhibitors of protein synthesis. This effect is associated with a reduction in KRAS levels, suggesting that the drugs may kill at least in part through reducing KRAS expression.

Overall, the link between codon usage and RAS expression is established through well-designed experiments in multiple cell lines, suggesting a general role for codon usage in regulating RAS expression. The observation that strongly-activating KRASQ61R mutations can drive resistance, despite low expression, while weakly-activating KRASG12/13 mutations require increased expression levels is very interesting, and suggests an novel mechanism of RAS-dependent drug resistance. However, the link between “translational upregulation” and increased KRAS levels is correlative. The manuscript would be greatly strengthened if the authors would provide evidence showing that the increased KRAS levels are the result of augmented protein synthesis, and are sufficient to confer drug resistance. Additionally, it is not clear if the lethality of drugs targeting the translational apparatus is due specifically to the decrease in KRAS levels, or due to either an overall decrease in protein synthesis or other targets affected by the compounds employed in the experiments. In principle, the manuscript is suitable for publication in Nature Communications, but these points should be addressed prior to acceptance.

Major comments

1. In Figure 5, the authors show a correlation between global protein synthesis rates and KRAS expression, but it is not clear if the relationship between the two is causal. The observation that mTOR inhibitors abrogate KRAS expression implicates increased translation initiation and/or elongation rates in driving KRAS expression. However, mTOR kinase phosphorylates many targets including components of the protein synthesis machinery, therefore a clearer mechanistic understanding of the causal effectors downstream of mTOR would strengthen the manuscript. For example, the authors could use a non-drug resistant cell line (i.e. LIM1215) and overexpress eIF4E and/or eEF2, to see if upregulation of a specific step in translation can increase KRAS levels and drive drug resistance.

We agree with Reviewer #2 that in addition to the correlation between global protein synthesis rates and KRAS expression, it would be beneficial to mechanistically define the effector downstream of mTOR that may be responsible for influencing KRAS expression. To address this point, we performed the experiment suggested by the reviewer, wherein we stably transduced a non-drug resistant cell line (LIM1215) with an empty vector construct or a construct expressing eIF4E, the key and limiting translational initiation component of the eIF4F complex. Overexpression of eIF4E caused an increase in KRAS protein levels (Supplementary Figure 5D). Through this experiment, we provide direct evidence that increasing this specific factor in translation initiation can increase KRAS expression. Interestingly, increased expression of KRAS, driven by eIF4E overexpression, was not sufficient to drive resistance to cetuximab in this non-drug resistant cell line (Supplementary Figure 5E). This latter result is likely due to the absence of a *KRAS* mutation in this cell line, which appears to be required for resistance. This observation is consistent with the finding that *KRAS* is frequently mutated in the setting of acquired cetuximab resistance.

2. In Figure 5, the link between global protein synthesis rate, cetuximab resistance, and KRAS expression is supported by data from only 2 cell lines. The authors should determine global protein synthesis rates in the panel of cell lines tested in Figure 5A, to see if the correlation between drug resistance and KRAS levels is a general trend.

We agree with Reviewer #2 that in the originally submitted manuscript, the link between global protein synthesis, cetuximab resistance, and KRAS protein expression was supported by data from only two cell lines, and that this should be expanded to additional cell lines. To determine global protein synthesis rates, puromycin labeling was performed in all six cell lines in Figure 5B. (We note that the data in Figure 5A was obtained from lysates that are available to us, however, we do not have all of these cell lines on hand. Thus, we performed western blotting and puromycin labeling on the subset of 6 of these lines that we have available.) We observed a statistically significant correlation between puromycin incorporation and KRAS protein expression in all six cell lines included in Figure 5B (Supplementary Figures 4A and 4B).

3. In Figure 5, it would be beneficial if the authors could provide an example of a gene not dependent on codon usage, to see if its expression is affected by “translational upregulation”. This would suggest that in the context of this cell line the effect of translational upregulation is specific to genes dependent on codon usage.

To address this question, we transduced the panel of 8 *KRAS* mutant cell lines presented in Figure 5C with a FLAG-tagged *NRAS* construct. *NRAS*, a member of the RAS family of proteins, has an equal representation of rare and common codons, and therefore is not considered to be strongly reliant on codon usage. We found no correlation between the expression levels of ectopic *NRAS* protein and ectopic *KRAS* protein, suggesting that the

increased protein translation we found to be transferrable to rare-codon containing constructs (Figures 5D and 5E, Supplementary Figure 4C) is not transferrable to genes whose expression levels are less dependent on codon usage (Supplementary Figures 4D and 4E). We have edited the manuscript text to accommodate this finding, which sheds important light on the translational upregulation observed in *KRAS* mutant lines by suggesting that this upregulation may be biased toward rare codon-enriched transcripts.

4. In Figure 6, it is not clear if treatment with translational inhibitors is killing the cells specifically through downregulation of *KRAS*, or through the inhibition of other targets. The authors should perform rescue experiments in the resistant cell lines by overexpressing *HRAS*, codon-optimized *KRAS*, and wild type *KRAS*. If the drugs kill through blocking the translational upregulation of *KRAS*, then expression of a *HRAS* or codon optimized *KRAS*, which are efficiently expressed, should rescue the killing effect of the translational inhibitors, whereas a wild type *KRAS* should not.

This is an insightful question and one that we also were curious to answer. As per Reviewer #2's suggestions, we performed rescue experiments in our cetuximab-resistant clone to determine whether overexpression of common codon-containing constructs, specifically *HRAS* and codon-optimized *KRAS* (*KRAS-COMMON*, or *KRAS-C*), would be able to rescue the selective killing effect of the translational inhibitors used in our experiments. Using this assay, we were unable to rescue the phenotype of resistant cells (Panel A below), as evidenced by no change in the half-maximal growth inhibition values for 4EGI-1. Importantly however, although we were able to overexpress the common codon constructs in this experiment (Panel B), their expression levels diminished to background levels following treatment with translational inhibitors (Panel C). This latter finding suggests that in the presence of these inhibitors, overexpression of both endogenous and ectopic *KRAS* protein remains suppressed, obscuring our interpretation of these data. As an alternative approach, we performed *KRAS* knockdown experiments using short hairpin RNAs (shRNAs) in both our parental and resistant cells, and found that loss of *KRAS* expression significantly affected cell viability by prolonging cell doubling time only in the cetuximab-resistant derivative (Supplementary Figures 5F and 5G). Combined with our findings that cetuximab-resistant cells develop collateral sensitivities to translational inhibitors (Figure 6D), which themselves suppress *KRAS* protein expression (Figure 6E), these data suggest that the hypersensitivity of resistant cells to translational inhibitors parallels the effects of *KRAS* knockdown. Nevertheless, other alternative explanations are possible, which we note in the text.

5. For Figure 6, it is not clear if “translational upregulation” is even occurring in these specific cell lines. An essential experiment would be to assay global protein synthesis rates in the parental and resistant cell lines.

This is an excellent point. To address this question, we performed puromycin labeling in the parental (P) and resistant clones (R1 and R2) presented in Figure 6A. Our data suggest that, as expected, the resistant clones demonstrate increased global protein translation (Supplementary Figure 5C).

6. Throughout the manuscript, conclusions based on western blots (e.g. Figs 1E, 2A, D, G, 3F, etc.) are often presented without quantification and statistical analysis. It is not clear if the results are representative of multiple experiments, or if the conclusions are only supported by what appear to be single experiments. Indeed, for any quality publication, multiple independent experiments should be performed in order to substantiate the authors’ claims.

We agree with Reviewer #2 and believe that multiple independent experiments should be performed prior to making any conclusions. We would like to clarify that the conclusions based on western blot data were made following multiple experimental replicates (in most cases 2 and in some cases 3), and across multiple cell lines and/or resistant derivatives, with replicate or triplicate experiments leading to the same conclusions in all cases. Similarly, the experiments performed during the revision process were repeated twice in all cases. We apologize for not making this clear in the original manuscript and have now included the number of replicates for each experiment in the figure legends.

Minor comments:

1. Figure 1D is not interpretable in its current state. The data should be presented in a clearer way.

In regards to this reviewer comment, we have increased the size of the font for each of the drug-treatment screens presented in Figure 5D. We hope that the general trend of *HRAS*^{G12V}-mediated resistance scoring higher (higher overall enrichment scores) across the screens presented is now more clear to readers, and that these data help to underscore our observation that even when both *HRAS*^{G12V} and *KRAS*^{G12V} scored in any particular screen, the enrichment score for *HRAS*^{G12V}-mediated resistance was always higher. However, if the reviewer has additional suggestions for how to more clearly present these data, we would certainly welcome them.

2. In Figure 6D, the authors switch between showing GI75 and GI50 values for different drugs. One or both values should be shown for all drug treatments.

Thank you for pointing out this oversight. In each case, we presented half-maximal or “true GI₅₀” values, which corresponded to GI₇₅ or GI₅₀ values depending upon the drug used. To avoid confusion moving forward, we are now reporting GI₅₀ values as half-maximal growth inhibition values throughout.

Thank you again for considering our manuscript, and please let me know if we can provide any additional or clarifying information.

REVIEWERS' COMMENTS:

Reviewer #2 (Remarks to the Author):

The authors have addressed my concerns, however prior to publication the authors should carefully revise the text of the manuscript.

Specifically:

-They should change "protein translation" to either protein synthesis or mRNA translation.

-Remove the statement that "eIF4E is a limiting factor in protein translation" as eIF4E is not limiting for global protein synthesis, and eIF4E loss or gain of function affects the translation of specific mRNAs.

-It would be beneficial for the readers if the authors in the discussion would add a paragraph on the general concept of how alterations in translational control lead to tumor development and drug response and how this fits with their new results.

Please find detailed point-by-point responses to each of the reviewers' final comments and suggestions.

Reviewer #2 (Remarks to the Author):

They should change “protein translation” to either protein synthesis or mRNA translation.

We thank Reviewer #2 for highlighting this key distinction, which points out the fact that proteins themselves are not translated; rather, the mRNA sequences encoding protein structure and function are the entities that are acted upon by the translational machinery. We have appropriately changed “protein translation” to protein synthesis where necessary.

Remove the statement that “eIF4E is a limiting factor in protein translation” as eIF4E is not limiting for global protein synthesis, and eIF4E loss or gain of function affects the translation of specific mRNAs.

We agree with the reviewer and maintain that eIF4E is a key factor involved in protein translation, but not a limiting factor in global protein translation. The statement has been modified to include the following language, “eIF4E, a key factor in protein synthesis”.

It would be beneficial for the readers if the authors in the discussion would add a paragraph on the general concept of how alterations in translational control lead to tumor development and drug response and how this fits with their new results.

We appreciate the reviewer's suggestion and have included the following paragraph to provide some context to our work:

“Previously, it has been demonstrated that oncogenic activation of translation initiation and/or elongation can support tumorigenesis by driving selective upregulation of specific mRNA transcripts⁴⁰. Indeed, seminal studies revealed that overexpression of eIF4E, involved in the translation initiation complex of protein synthesis, was sufficient to drive tumorigenesis in cell lines and spontaneous tumorigenesis in mice⁴¹⁻⁴³. Additionally, deregulation of translation has more recently been identified as both a primary downstream consequence of oncogenic signaling, as well as a central mediator of resistance to clinical therapies, including drugs targeting the MAPK and PI(3)K-AKT-mTOR signal transduction pathways^{44,45}. These findings, along with data showing that cancer cells co-opt translational machinery to support tumor growth, have driven increased interest in targeting translational control as a method to selectively kill cancer cells⁴⁶.”